# Multiscale model of defective interfering particle replication for influenza A virus infection in animal cell culture

**Daniel Rüdiger**[1☯]*, **Lars Pelz**[1☯], **Marc D. Hein**[2], **Sascha Y. Kupke**[1]*, **Udo Reichl**[1,2]

**1** Max Planck Institute for Dynamics of Complex Technical Systems, Magdeburg, Germany, **2** Chair of Bioprocess Engineering, Institute of Process Engineering, Faculty of Process & Systems Engineering, Otto-von-Guericke University, Magdeburg, Germany

☯ These authors contributed equally to this work.
* ruedigerd@mpi-magdeburg.mpg.de (DR); kupke@mpi-magdeburg.mpg.de (SYK)

**Data Availability Statement:** The experimental data are within the manuscript and its Supporting Information files. The code for model simulation

## Abstract

Cell culture-derived defective interfering particles (DIPs) are considered for antiviral therapy due to their ability to inhibit influenza A virus (IAV) production. DIPs contain a large internal deletion in one of their eight viral RNAs (vRNAs) rendering them replication-incompetent. However, they can propagate alongside their homologous standard virus (STV) during infection in a competition for cellular and viral resources. So far, experimental and modeling studies for IAV have focused on either the intracellular or the cell population level when investigating the interaction of STVs and DIPs. To examine these levels simultaneously, we conducted a series of experiments using highly different multiplicities of infections for STVs and DIPs to characterize virus replication in Madin-Darby Canine Kidney suspension cells. At several time points post infection, we quantified virus titers, viable cell concentration, virus-induced apoptosis using imaging flow cytometry, and intracellular levels of vRNA and viral mRNA using real-time reverse transcription qPCR. Based on the obtained data, we developed a mathematical multiscale model of STV and DIP co-infection that describes dynamics closely for all scenarios with a single set of parameters. We show that applying high DIP concentrations can shut down STV propagation completely and prevent virus-induced apoptosis. Interestingly, the three observed viral mRNAs (full-length segment 1 and 5, defective interfering segment 1) accumulated to vastly different levels suggesting the interplay between an internal regulation mechanism and a growth advantage for shorter viral RNAs. Furthermore, model simulations predict that the concentration of DIPs should be at least 10000 times higher than that of STVs to prevent the spread of IAV. Ultimately, the model presented here supports a comprehensive understanding of the interactions between STVs and DIPs during co-infection providing an ideal platform for the prediction and optimization of vaccine manufacturing as well as DIP production for therapeutic use.

and optimization is available at https://github.com/
ModIAV/STV_DIP_Coinfection.

**Funding:** This work was partially supported by the
Defense Advanced Research Project Agency
INTERCEPT program (https://www.darpa.mil/
program/intercept) under Cooperative Agreement
W911NF-17-2-0012 (LP). The funders had no role
in study design, data collection and analysis,
decision to publish, or preparation of the
manuscript.

**Competing interests:** I have read the journal's
policy and the authors of this manuscript have the
following competing interests: A patent for the use
of OP7 (a DIP containing point mutations instead
of a deletion in viral genome segment 7) as an
antiviral agent for treatment of IAV infection is
pending. Patent holders are S.Y.K. and U.R.
Another patent for the use of DI244 and OP7 as an
antiviral agent for treatment of coronavirus
infection is pending. Patent holders are S.Y.K., U.R.
and M.D.H. The remaining authors declare no
conflict of interest.

## Author summary

Influenza viruses replicate inside their hosts after infection. Along with the release of wild-type standard virus (STV), they can also generate specific kinds of particles that have deletions in their genome. These so-called "defective interfering particles" (DIPs) are unable to replicate on their own. However, during co-infection with STV they can interfere with the viral life cycle suppressing STV production and instead release a large number of progeny DIPs. Recent studies have shown promising results regarding their potential as antiviral agents. To characterize the interactions between STVs and DIPs during co-infections, we infected animal cell cultures using 12 different multiplicities of infection for STVs and DIPs, respectively. We measured intra- and extracellular infection dynamics and show that STV replication can be suppressed completely using high amounts of DIPs. Then, we developed a mathematical model that describes co-infection dynamics closely using a single set of parameters. We used this model to predict optimal dosing ratios for DIPs to suppress STV infections and for cell culture-derived production of DIPs for antiviral therapy.

## Introduction

Infectious diseases continue to pose significant and unpredictable risks for human and animal life despite enormous preventive and therapeutic efforts taken over the last 100 years. The current COVID-19 pandemic clearly demonstrates that newly emerging viruses can lead to millions of deaths and severe impacts on the global economy [1]. Influenza A virus (IAV) has the potential to produce equally dangerous epi- and pandemics due to its high mutation rate and a natural reservoir in various species [2,3]. Prevention and treatment strategies focus mostly on vaccination and the administration of antiviral drugs. However, emerging resistances to current antivirals may limit treatment approaches [4,5], which emphasizes the need for an improvement and expansion of the therapeutic catalogue.

Defective interfering particles (DIPs) are structurally similar to their corresponding standard virus (STV), but replication-incompetent due to a large internal deletion in at least one of their eight viral genome segments [6–10]. During co-infection with the STV, which acts as a helper virus by providing the missing genomic information, they can generate progeny DIPs as well as reduce production and release of STV particles strongly. The decrease in STV production was theorized to be related to a growth advantage of the defective interfering (DI) genome over its full-length (FL) counterpart [7,11,12]. Currently, the shorter length of DI RNA is a prominent hypothesis for the source of this advantage [7], however, the underlying mechanisms are still not understood completely. Besides IAV, nearly all RNA viruses produce DIPs [8,13]. Based on their inhibiting effect during co-infection, DIPs are considered promising candidates for antiviral therapy. Previous animal studies showed that the administration of DIPs could successfully prevent and treat IAV infections in mice and ferrets [14–18]. However, the exact mechanisms of DIP interference, the structural requirements for a potent inhibition of IAV infection and the selection of an optimal dose for therapeutic application remain elusive. In addition, the design and optimization of the production of DIP preparations for manufacturing of antivirals is a challenge. Therefore, a comprehensive mathematical model describing the interplay between STVs and DIPs in cell culture during co-infection on a systems level could support further research into these areas.

Previous model-based studies of DIP infection examined mostly *in vitro* virus propagation on the cell population level [19–22]. However, the competition for cellular resources and inhibition of STV production and release occurs during intracellular virus replication. While few

studies also focused specifically on the effect of DIPs on the intracellular level [23,24], the spreading of DIPs in cell populations was not taken into account. Finally, some previously developed models considered both levels of infection [25,26]. However, these contained rather basic representations of the intracellular virus replication dynamics. A general outcome of model-based DIP infection studies was the impact of the ratio between STVs and DIPs for inhibition, i.e., the applied multiplicity of infection (MOI) and multiplicity of DIPs (MODIP).

Besides DIP production-related issues, a better understanding of the interaction between these two virus populations could also contribute to the development of innovative therapeutic approaches. Applying different MOIs and MODIPs for an infection of animal cells can influence the dynamics of virus propagation significantly. Using vast amounts of both STVs and DIPs leads to more co-infections, which favor DIP production. In contrast, for low MOI and MODIP conditions, only few co-infections occur and, consequently, the STVs can replicate unhindered and overcome inhibition by DIPs. In addition, the MOI can change drastically during the course of an infection as more and more virions are released [27], which is expected to also apply to the MODIP. Therefore, the interplay between the infection conditions on the cell population level and the resulting effects on intracellular replication could be a key factor to understand DIP inhibition dynamics.

To elucidate the complex interactions between STVs and DIPs, we conducted a comprehensive set of experiments using various combinations of MOI and MODIP, and analyzed the infection dynamics on the intracellular and population level in Madin-Darby Canine Kidney suspension (MDCKsus) cells. In addition, we developed a multiscale model for STV and DIP co-infection on the cellular and the population level. As a starting point we considered two previously published models; one focusing on virus replication at the single-cell level [23], the other describing STV replication and propagation for different MOIs [27]. We introduced populations of cells infected only by DIPs and co-infected cells on the population level as well as specific viral mRNA regulations and viral genomic RNA synthesis suppression dynamics on the intracellular level. Then, we utilized the obtained experimental data to calibrate the multiscale model and to predict the impact of different MOI/MODIP conditions on the infection dynamics.

## Results

### High DIP loads suppress STV infection and prevent virus-induced apoptosis

To examine how a variation of MOI and MODIP affects the overall replication dynamics during STV and DIP co-infection, we infected MDCKsus cells with different combinations of STV (A/PR/8/34, H1N1) and purely clonal DIP input. We selected MOIs of $10^{-3}$, 3, 30 and MODIPs of 0, $10^{-3}$, 3, 30 to create 12 different infection conditions (Fig 1A). For our experiments, we used a prototypic, well-characterized DIP referred to as DI244 [28]. DI244 particles contain a deletion on the virus genome segment 1 encoding for polymerase basic protein 2 (PB2). The applied MOIs and MODIPs ranged over four orders of magnitude to cover highly different infection scenarios.

For all 12 infection conditions (Fig 1A), we measured the dynamics of viral mRNA and genomic viral RNA (vRNA) on the intracellular level (Figs 2 and S5–S10) using real-time reverse transcription qPCR (real-time RT-qPCR). On the extracellular level, we determined the yield of infectious particles (quantified by plaque assay) and the total yield of STVs and DIPs (quantified by real-time RT-qPCR). Furthermore, we measured the fraction of infected and apoptotic cells using imaging flow cytometry (S3 and S4 Figs).

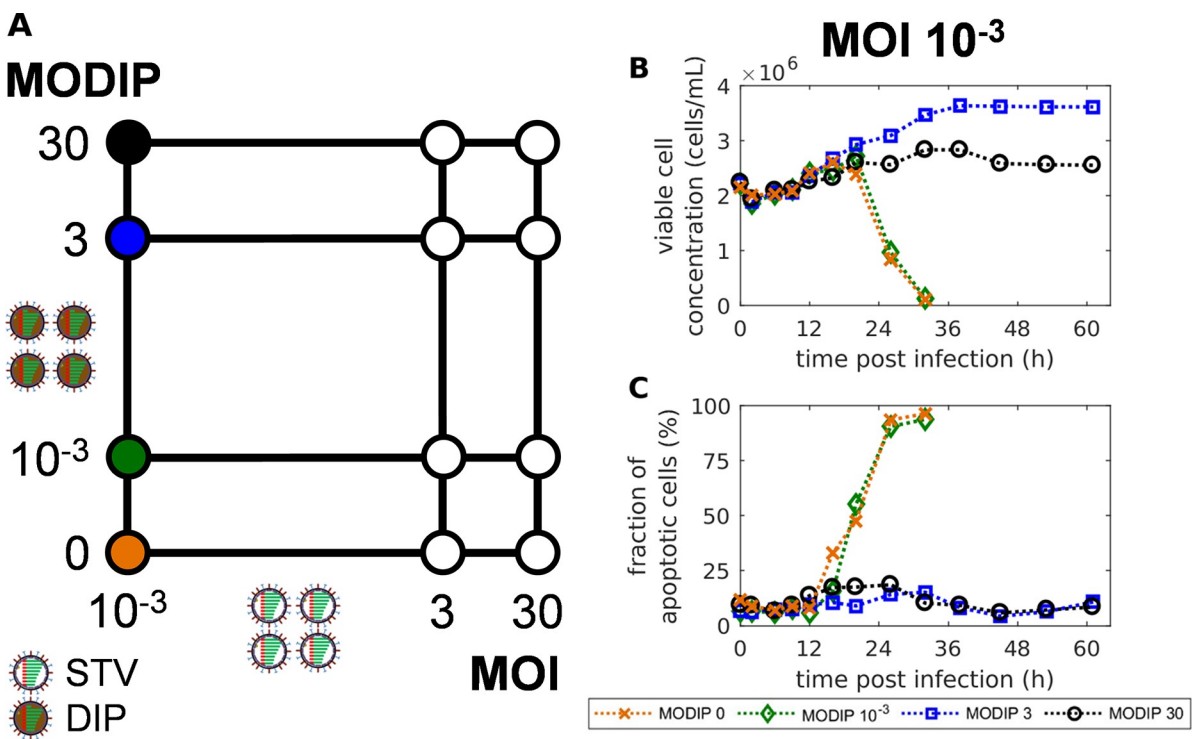

**Fig 1. Addition of defective interfering particles (DIPs) can prevent virus-induced apoptosis and protect MDCKsus cells from standard virus (STV) infection.** (A) Schematic depiction of the 12 different MOI and MODIP conditions used for infection experiments. (B) Viable cell concentration and (C) the fraction of apoptotic cells for infections with MOI $10^{-3}$ and MODIPs of 0, $10^{-3}$, 3 and 30. Results for all infection conditions are shown in S1 Fig.

Our experimental results show that MDCKsus cells were protected from virus-induced cell death when infected at a low MOI of $10^{-3}$ combined with MODIPs of 3 and 30 (Fig 1B). These two infection conditions will from here on be referred to as L3 (MOI $10^{-3}$ + MODIP 3) and L30 (MOI $10^{-3}$ + MODIP 30). For L3, the cells continued to grow until 38 hours post infection (hpi), while for L30 less cell growth occurred which seemed to be affected by the higher concentration of DIPs (Fig 1B). In contrast, all other infection conditions lead to virus-induced apoptosis and cell death after addition of STVs (S1 Fig). Interestingly, for L3 and L30 conditions, the apoptosis level stayed below 20% (Fig 1C) although the cells were infected by high amounts of DIPs. This is likely caused by a significant reduction of intracellular virus replication, especially the low accumulation of vRNAs (Fig 2), whereby apoptotic processes are not triggered.

Both infectious and total virus titers were reduced significantly for L3 and L30 (S3 Fig). Virus replication still occurred on a low level, however, compared to the other conditions a reduction of the infectious virus titer at 26 hpi by five to six orders of magnitude was observed in L3 and L30, respectively.

In sum, the addition of high amounts of DIPs during a STV infection at a low MOI can protect cells from virus-induced cell death and reduce viral titers severely.

## DIP infection leads to considerable viral mRNA transcription even in the absence of co-infections

On the intracellular level, we focused on the dynamics of viral RNAs. Negative-sense vRNAs contain the genomic information and enter the cellular nucleus during infection. There, they

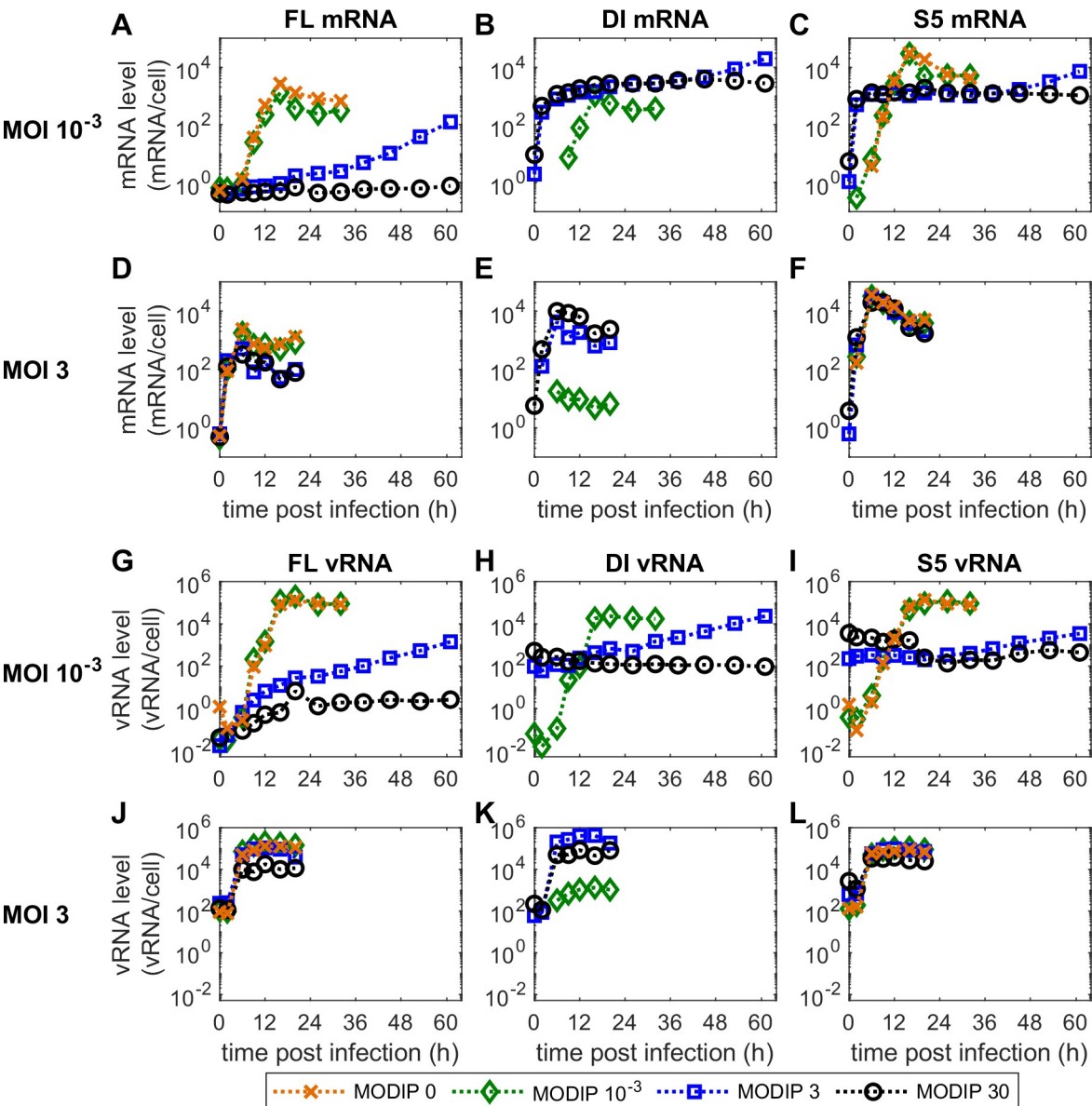

**Fig 2. Viral RNA dynamics in infected MDCKsus cells are strongly influenced by the applied concentrations of STVs and DIPs.** Real-time RT-qPCR measurements of (A-F) viral mRNA and (G-L) vRNA levels for different infection conditions. The results for (left) FL segment 1, (middle) DI segment 1 and (right) FL segment 5 are shown. Results for the remaining infection conditions are presented in S5–S7 Figs.

act as templates for the synthesis of two positive-sense RNAs, i.e., viral mRNA and complementary RNA (cRNA). The former is required to perform viral protein translation and the latter is a replicative intermediate that can be transcribed to produce progeny vRNAs. We measured the levels of vRNA and viral mRNA for three different genome segments, i.e., the FL segment 1, the truncated DI244 segment 1 and segment 5 (S5).

The maximum levels of viral mRNA show clear differences between the three measured segments. S5 mRNA reaches the highest levels and FL mRNA levels are reduced by over one order of magnitude (Fig 2). DI mRNA achieves levels between the other two segments. This

implies that there is a fundamental mechanism that regulates the mRNA accumulation of different segments as this behavior is replicated over all infection conditions (S5–S7 Figs).

The levels of viral mRNA are reduced strongly for the low MOI, high MODIP conditions L3 and L30. FL mRNA is not produced in L30 (Fig 2A) and increases slowly over time in L3. Interestingly, DI and S5 mRNA initially accumulate to similar levels as for other infection conditions using high MOI or MODIP until 2 hpi. Then, their increase stagnates and they stay at this level until at least 45 hpi. Due to the low MOI and high MODIP for conditions L3 and L30, the vast majority of cells were infected only by DIPs and not by STVs. Therefore, the observed levels of DI and S5 mRNA were generated by DIP-only infected cells and did not require a co-infection by STVs providing missing genomic information.

The accumulation of vRNA for all three observed segments is suppressed completely for L30 but shows a slow and steady increase for L3 (Fig 2). Because of the defective genome in DI244, PB2 cannot be produced in cells only infected by DIPs. PB2 is a subunit of the viral RNA-dependent RNA polymerase (RdRp) which is essential for virus replication. Therefore, in cells only infected by a DIP and no STV, the synthesis of progeny vRNA is prevented. For the other conditions, FL and S5 vRNA show typical accumulation dynamics. However, both vRNA and viral mRNA trend towards reduced levels with increasing MODIPs.

Taken together, the DI244 used in our experiments is capable of viral mRNA transcription at considerable levels without the STV functioning as a helper virus.

## Developing a multiscale model of STV and DIP co-infection

The mathematical model presented comprises basic aspects of two of our previously published models but expands their scope by including viral RNA replication phenomena observed for DIP propagation in this study. The model structure is based on an MOI-sensitive multiscale model of STV infection in animal cell culture [27]. For the description of the data obtained in our experiments, we expanded this model by considering the dynamics of DIPs on the cell population level and the intracellular level using an adapted model of intracellular DIP replication [23].

To this end, we introduced cells only infected by DIPs ($I_{\mathrm{DIP}}$) and co-infected cells ($I_{\mathrm{CO}}$) to the existing uninfected ($T$), STV-only infected ($I_{\mathrm{STV}}$), uninfected apoptotic ($T_A$) and infected apoptotic cells ($I_A$) on the population level (Fig 3, Eqs (11)–(16)). We assumed that DIP-only infected cells are incapable to produce progeny virions due to missing genomic information and, therefore, neglected virus entry, replication and release on their intracellular level. On the other hand, co-infected cells release both progeny STVs and DIPs. In accordance with the description of STV-only infected cells (Eq (13)), co-infected cells were also represented as an age-segregated population $I_{\mathrm{CO}}(t, \tau)$ (Eq (14)). Additionally, DIPs themselves ($D$) as well as their different binding and endocytosis states, i.e., attached DIPs ($D_n^{\mathrm{Att}}$) and DIPs in endosomes ($D^{\mathrm{En}}$), were implemented analogous to STV particles (Eqs (S59)-(S69)).

On the intracellular level the DIP entry and nuclear import, the replication of the DI segment and the release of progeny DIPs were considered. We modified the model structure slightly in accordance with specific assumptions used in the original multiscale model [27]. Therefore, we modified the STV and DIP release kinetics by including a maximum release rate of infected cells (Eqs (7)–(10)).

The fraction of infectious virions released (FIVR, $F_{\mathrm{Par}}$) by infected cells was crucial in the development of the original multiscale model as it determined how rapidly an infection can spread when low MOI concentrations are used [27]. In the original model, the FIVR had to be adjusted for application for different infection conditions as it showed a reduced value for high MOI ($F_{\mathrm{Par}} = 0.034$) compared to low MOI scenarios ($F_{\mathrm{Par}} = 0.26$). We speculated that this

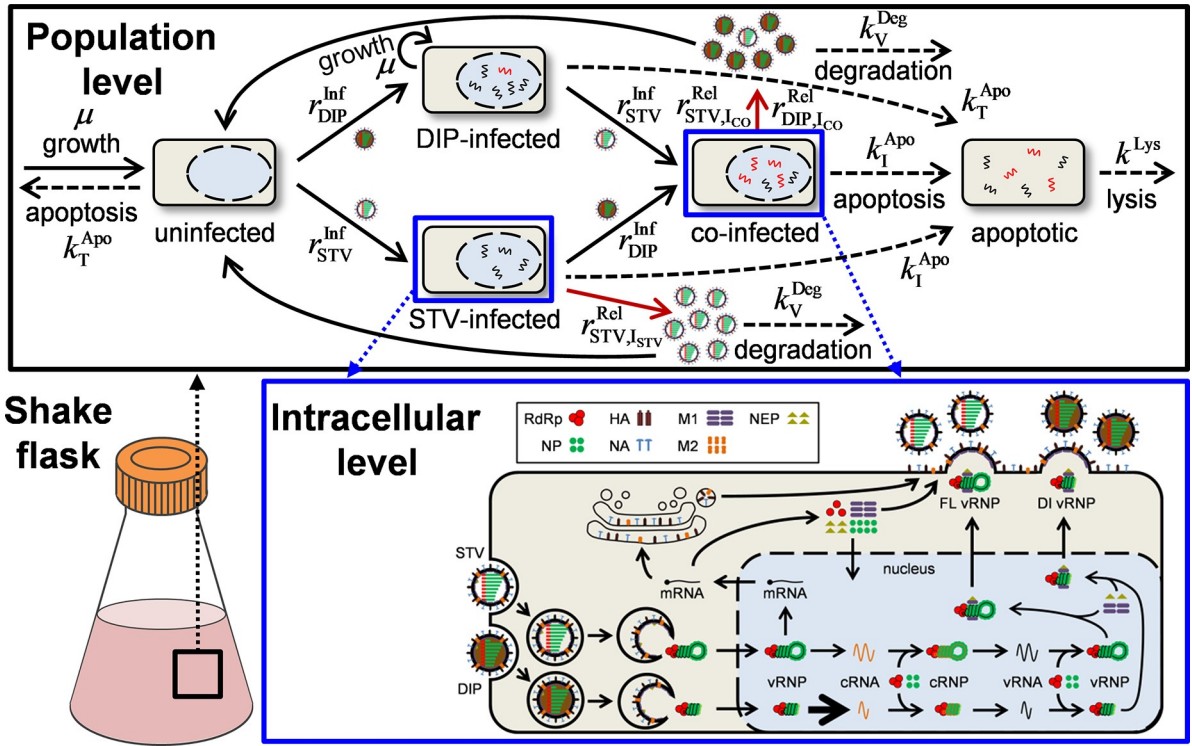

**Fig 3. Schematic depiction of the multiscale STV and DIP co-infection model.** (Top) The population level of infection describes growth and apoptosis of uninfected cells, their infection by either STVs or DIPs, the co-infection of STV-only infected and DIP-only infected cells, virus-induced apoptosis of all infected cells and the lysis of apoptotic cells. STVs are released from STV-only infected and co-infected cells, DIPs are only released from co-infected cells and both are cleared via virus degradation. (Bottom) Virus entry, nuclear import, viral RNA and protein synthesis, nuclear export and progeny virion release in STV-only infected cells and co-infected cells is simulated using the same intracellular model. Figure adapted from [29].

variation was most likely induced by DIPs, which affect STV replication more strongly in high MOI conditions. For the development of our model of STV and DIP co-infection, we applied a single value for the FIVR, because the impact of DIP interference was implemented in the model itself.

To take into account the specific impact of MOI and MODIP conditions as well as their dynamics over time, we consider the current virus concentrations explicitly when calculating intracellular dynamics. In the original multiscale model, the initial conditions for the simulation of the intracellular level were based solely on the MOI at time of infection. For the model developed in this study, we use varying initial conditions, which are adapted to the current concentrations of STVs and DIPs in the cell culture to simulate intracellular dynamics. While this increases computational burden, it considers the dynamic changes of MOI and MODIP during infection.

The original multiscale model [27] relied on experimental data obtained from a different cellular system (adherent MDCK cells). The seed virus used in this study, however, was adapted to MDCKsus cells that show overall faster infection dynamics. Therefore, a re-evaluation of various process parameters that may have been affected by this change in the cell line was required. Additionally, the model was adapted to utilize infectious STV titer measurements obtained via plaque assay (PFU/mL) instead of the previously used tissue culture infection dose (TCID$_{50}$) assay [30] that typically results in higher titers.

The basic model of STV and DIP co-infection was calibrated to the intra- and extracellular data of 12 infection conditions (Figs 1 and 2). However, for most infection conditions, model simulations showed large deviations to the measured values (S2 Fig). Especially for low MOI conditions, large deviations for the observed intracellular properties were apparent (S11 Fig). Therefore, we were not able to obtain a set of parameters describing all measured dynamics simultaneously. Most likely, this is due to the inherent complexity of the interaction of STVs and DIPs during infection.

In sum, we established a mathematical multiscale model of STV and DIP co-infection by taking into account DIP replication and spreading on the intracellular and cell population level, respectively, to describe the infection dynamics observed in our experiments (Fig 1). However, estimating a set of parameters that describes all infection conditions could not be achieved with this basic model.

## Extension of the basic model of STV and DIP co-infection

To address the observed discrepancies between our basic model and the measured dynamics during STV and DIP co-infection in animal cell culture, we implemented several targeted changes to the model equations.

First, we wanted to address the discrepancies observed between the levels of the three different viral mRNAs (Fig 4A). Although the overall dynamics of viral mRNA accumulation could be captured, the levels of FL and DI mRNA were overestimated, while the levels of S5 mRNA were underestimated. Previously, a clear distinction was postulated for segments encoding for proteins of the viral polymerase RdRp, i.e., segment 1 to 3, and the other segments 4 to 8 [31–33]. The polymerase segments seemed to accumulate to significantly lower levels. Therefore, we decided to implement this effect by a simple parameter $f_M$ that reduces mRNA transcription in polymerase segments including the DI segment (Eq (2)). This clearly improved the description of our experimental data compared to the basic model and enabled the representation of the different levels of accumulated viral mRNA (Figs 4B and S5–S7).

Another significant deviation between the simulation of the basic model and the experimental data could be observed for the levels of DI and S5 mRNA in low MOI, high MODIP conditions L3 and L30 (Fig 4C). These two mRNAs still accumulated to considerable numbers, although nearly all infected cells should have been infected with a DIP but no STV as a helper virus. We assumed initially that cells just infected by DIPs do not produce any viral RNAs. Yet, previous IAV infection studies showed that the "primary transcription" of viral mRNAs by incoming parental vRNAs can lead to significant levels [34–36].

To implement this hypothesis, we used a modified version of the intracellular equation describing viral mRNA kinetics (Eq (1)) for DIP-only infected cells. In this simplified equation (Eq (3)), the negative feedback induced by RdRp is removed, because cells only infected by DI244 cannot synthesize functional PB2, which is essential for RdRp formation. Furthermore, the primary transcription now depends on the raw input of viral ribonucleoprotein (vRNP) templates per cell from the initial infection. Using this simple description, we can capture the level of viral mRNA accumulation in L30 closely (Figs 4D and S5–S7). For L3, the combination of primary transcription and the regular viral mRNA generation in co-infected cells also describes the initial plateau and the following increase.

Then, we focused on the vRNA dynamics, which were not represented completely (S11 Fig). Therefore, we fixed every model parameter except $k_V^{\text{Syn}}$, which describes the rate of vRNA synthesis, and calibrated the model to the experimental data. Thus, we identified that $k_V^{\text{Syn}}$ was estimated to very similar values for low initial DIP concentrations, i.e., MODIP 0 and $10^{-3}$, but showed a clear reduction when MODIP 3 and 30 were used for infection (Fig 4E). Specifically,

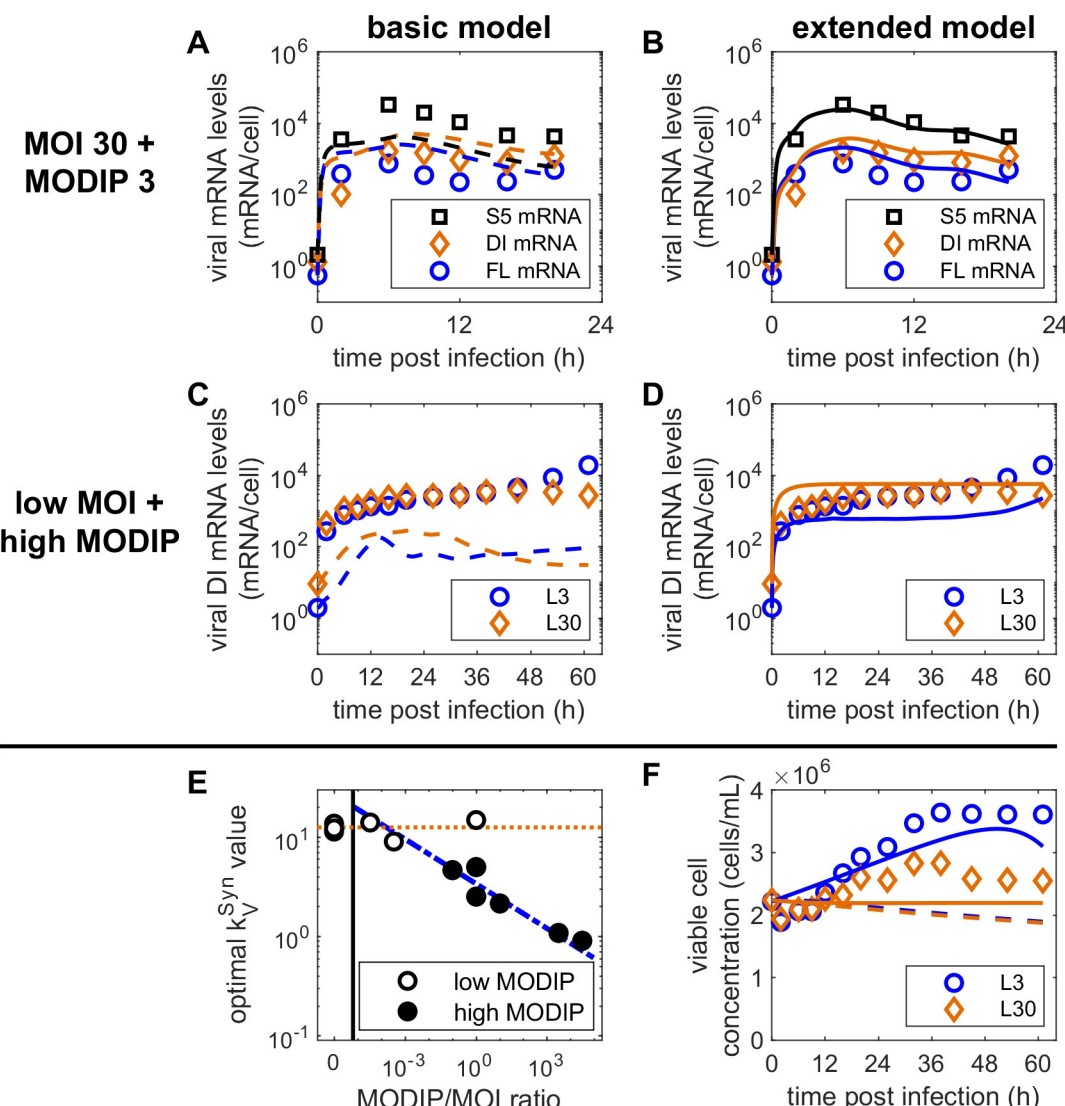

**Fig 4. Adaptations of viral RNA synthesis and cell growth mechanisms of the basic model improve the description of observed infection dynamics.** (A+C+F) Dashed lines show model simulations of the basic model fitted to experimental data, (B+D+F) solid lines depict simulations of the extended model calibrated to the same measurements. (A+B) Dynamics of viral mRNA for segment 5, full-length segment 1 and defective interfering segment 1 using MOI 30 and MODIP 3. (C+D) Accumulation of viral DI mRNA for conditions L3 (MOI $10^{-3}$ and MODIP 3) and L30 (MOI $10^{-3}$ and MODIP 30). (E) Estimated values for the parameter $k_V^{Syn}$ describing vRNA synthesis for different infection scenarios based on the applied MODIP-to-MOI ratio. The orange dotted line depicts the average $k_V^{Syn}$ value for low MODIP infections (empty circles) and the blue dash-dotted line represents the dependency of $k_V^{Syn}$ on the MODIP-to-MOI ratio for high MODIP conditions (full circles). The vertical black line separates infections only using STVs from infections with MODIP > 0. (F) Dynamics of the viable cell concentration for low MOI, high MODIP conditions L3 and L30. Experimental data for all other infection conditions are shown in S1 and S3–S10 Figs.

we observed a direct relation of the parameter value to the applied ratio of MODIP to MOI. Consequently, we introduced a dependency of the parameter $k_V^{Syn}$ on the MODIP-to-MOI ratio used during infection (Eq (4)). Fortunately, this modification to the model did not only enable the description of STV and DIP co-infection for all conditions using a single set of parameters, but also improved the description of our experimental data considerably (S2–S10 Figs).

Additionally, we considered that the cell growth observed for low MOI, high MODIP conditions L3 and L30 seemed to be lower with increased MODIP (Figs 1B and S1). Therefore, we introduced a factor $f_\mu$ that reduces the specific cell growth rate during infection depending on the initial DIP concentration (Eq (19)). While the fraction of apoptotic cells did not increase when infected with a large quantity of DIPs (Fig 1C), they nevertheless showed an impaired cell growth. By using this additional factor, we were able to describe the differing growth dynamics for conditions L3 and L30 (Fig 4F).

Finally, the extended model of STV and DIP co-infection (Eqs (S1)-(S74)) was fitted to measurements from 12 different combinations of MOI and MODIP conditions (Table 1). Model simulations capture experimental data on the intracellular and cell population level closely (Figs 5 and S3–S10). Especially the effects of STV suppression for low MOI, high MODIP conditions L3 and L30 can be described well (Fig 5L). On the intracellular level, the balance between vRNA and viral mRNA can be captured for nearly all conditions. Furthermore, viral titers and cell population dynamics are described well. The extended model comprises 132 ODEs and 73 parameters (basic model: 130 ODEs, 68 parameters). For 8 out of 12 experiments fitted, the extended model showed lower values for the Akaike information criterion (S1 Table) and is, therefore, preferable [37]. This applies, in particular, for MOI $10^{-3}$ combined with a low MODIP, where the experimental data could not be described using the basic model (S11 Fig). Furthermore, for high MOI combined with high MODIP conditions the data is fitted better by the extended model. Overall, while the basic model displays a certain advantage to describe some infection conditions, the extended model is able to capture all conditions simultaneously.

In summary, we extended our basic model of STV and DIP co-infection by considering (I) segment-specific viral mRNA production, (II) the primary transcription of viral mRNA in DIP-only infected cells, (III) a reduction of vRNA synthesis depending on the applied

**Table 1. Parameters estimated from the experimental data in S1 and S3–S10 Figs.**

| Parameter | Value | Confidence interval (95%)[a] |
|---|---|---|
| $F_{\mathrm{Par}}(0)$ (−) | $3.6\times10^{-3}$ | $(0.3\text{–}48.9)\times10^{-3}$ |
| $F_{\mathrm{Adv}}$ (−) | 0.32 | 0.07–0.84 |
| $F_M$ (−) | 0.12 | 0.006–0.53 |
| $F_\mu$ (−) | 0.63 | 0.2–1 |
| $k_T^{\mathrm{Apo}}$ (h$^{-1}$) | $1.18\times10^{-2}$ | $(0.1\text{–}1.3)\times10^{-2}$ |
| $k^{\mathrm{Fus}}$ (h$^{-1}$) | 58.3 | 9.5–258.8 |
| $K_I$ (h$^{-1}$) | 0.27 | 0.05–0.35 |
| $k^{\mathrm{Lys}}$ (h$^{-1}$) | 0.16 | 0.02–0.5 |
| $K_R$ (molecules) | $7.8\times10^{3}$ | $(1.1\text{–}30.9)\times10^{3}$ |
| $k_{\mathrm{Red}}^{\mathrm{Rel}}$ (h$^{-1}$) | $4.1\times10^{-4}$ | $(0.7\text{–}16.1)\times10^{-4}$ |
| $k^{\mathrm{Rel}}$ (virions · h$^{-1}$) | $6.15\times10^{3}$ | $(0.9\text{–}19.3)\times10^{3}$ |
| $K_V$ (h$^{-1}$) | 20.1 | 4.7–78.5 |
| $K_{V^{\mathrm{Rel}}}$ (virions) | 1.8 | 0.3–6.8 |
| $\tau_{\mathrm{Apo}}$ (h) | 6.65 | 5.0–18.0 |
| $\nu_1$ (−) | 5.2 | 2.0–47.7 |
| $\nu_2$ (−) | 0.1 | 0.002–0.23 [b] |

[a] 95% confidence intervals were obtained from the Q0.025 and Q0.975 quantiles of 1250 bootstrap iterations [38].
[b] Estimates reached lower bootstrap parameter bounds.

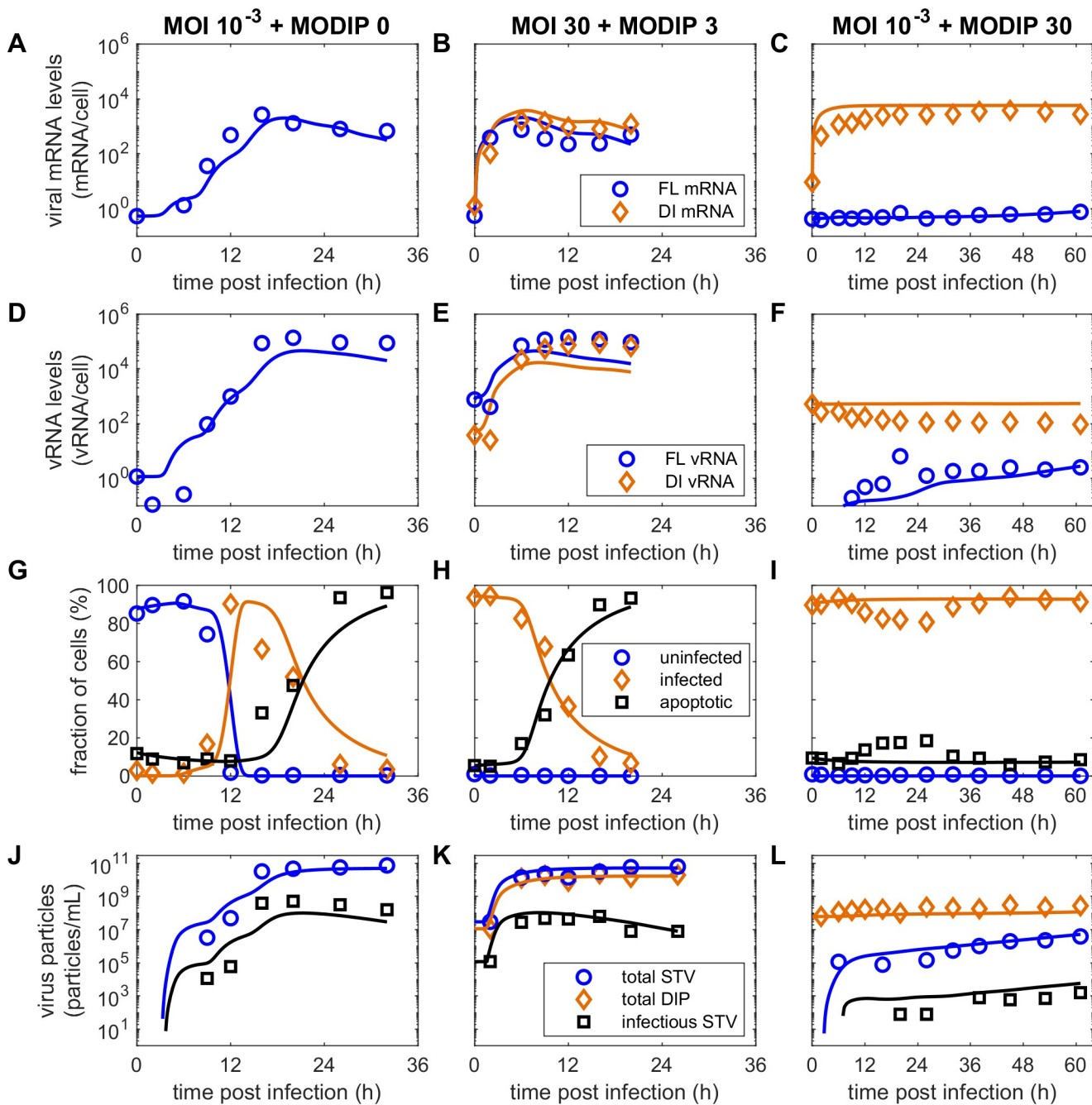

**Fig 5. The extended multiscale model captures infection dynamics on the intracellular and cell population level for all measured infection conditions.** Curves depict simulations of the extended model fitted to (A-C) cell-specific viral mRNA, (D-F) cell-specific vRNA, (G-I) cell population and (J-L) extracellular virus titers measured in MDCKsus cell cultures infected with different amounts of influenza A/PR/8/34 (H1N1) and defective interfering particles (DI244). Results for MOI $10^{-3}$ and MODIP 0 (first column), 30 and 3 (second column), and $10^{-3}$ and 30 (third column) are shown. The figures presenting cell population dynamics (G-I) show fits to uninfected non-apoptotic, infected non-apoptotic, as well as the sum of uninfected and infected apoptotic cells. The extended model is based on Rüdiger et al. [27] and Laske et al. [23], but additionally considers primary transcription of viral mRNA, segment-specific viral mRNA production, a reduced vRNA synthesis for high MODIP conditions and a DIP-induced reduction of cell growth. Simulation results for all other infection conditions are shown in S3–S10 Figs.

MODIP-to-MOI ratio, and (IV) DIP-induced cell growth reduction. The extended model describes all examined infection conditions using a single set of parameters.

## Model simulations predict the ratio of MODIP to MOI required to reduce infectious STV titers significantly

In a next step, the extended model was used to predict the optimal infection conditions for the successful suppression of STV propagation and the generation of large DIP quantities. These two scenarios are especially relevant regarding the production of DIPs for antiviral therapy and their potential application against STV infection.

Generally, high doses of DIPs are required for strong inhibition of STV infection (Fig 6A). However, also low doses of DIPs can show an inhibiting effect when the MOI is significantly lower than the MODIP. Our simulations predict that using ratios of MODIP to MOI of 1:1 enables a reduction of infectious STV titers by a factor of 10 compared to a DIP-free infection. To induce a strong reduction of infectious virus titers, i.e., by at least four orders of magnitude, a ratio of $10^4$:1 is required.

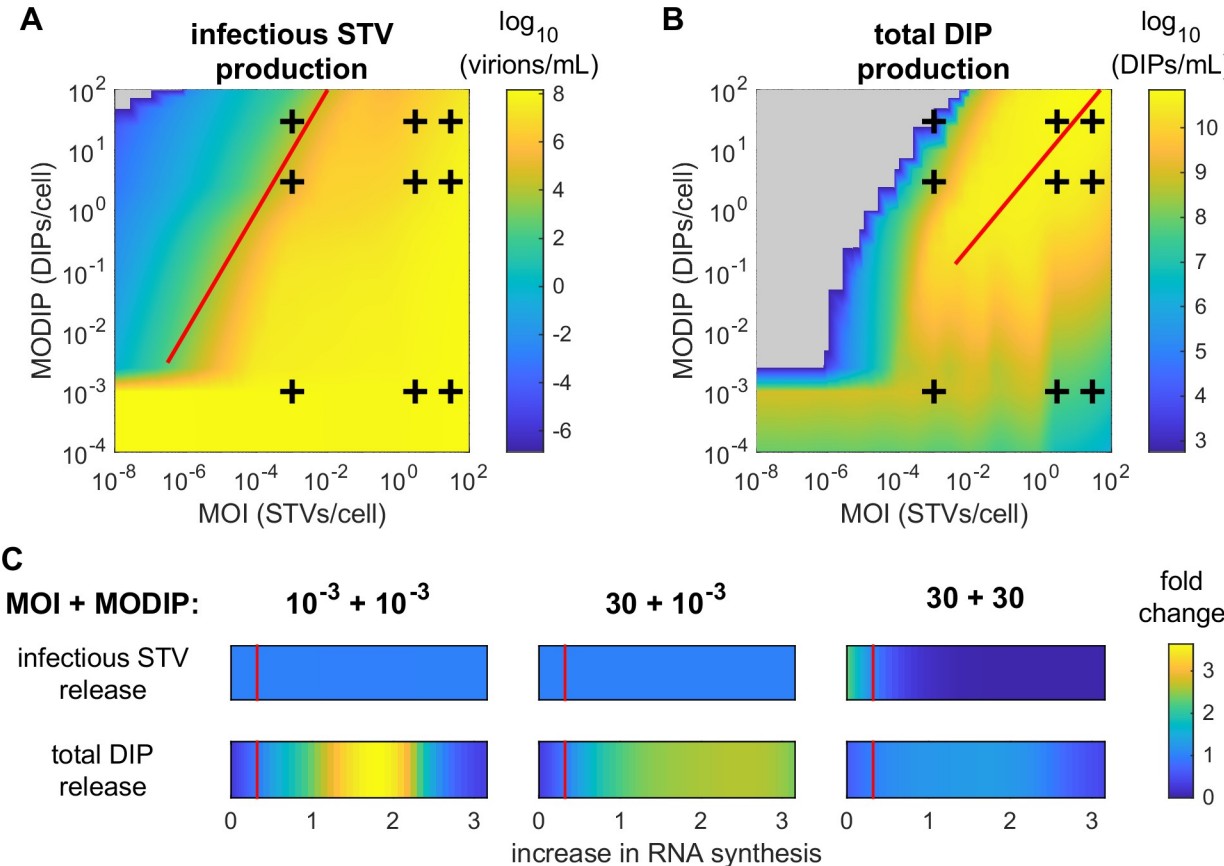

**Fig 6. STV infection suppression and DIP production are strongly affected by the infection conditions.** (A+B) Infections of MDCKsus cells were simulated with the extended model using MOIs and MODIPs in the range of $10^{-8}$–$10^2$ and $10^{-4}$–$10^2$, respectively. The predicted concentrations of (A) infectious STVs and (B) total DIPs at 48 hpi determine the color of the heat map. The experimentally observed infection scenarios (+) are depicted. The solid red line indicates (A) an MODIP-to-MOI ratio of $10^4$:1 and (B) the optimal multiplicity ratios for DIP production. For the switch between regular and reduced vRNA synthesis for low and high initial DIP concentrations a threshold value of $F_{\text{MODIP}}$ = $10^{-3}$ was used. Grey areas indicate that no production of either STVs or DIPs did occur. (C) The predicted fold-change for yields of infectious STVs and total DIPs at 24 hpi depending on a reduction or increase in the replication advantage of DI cRNAs over their FL counterpart is presented. The parameter $F_{\text{Adv}}$ was varied in the range of 0 to 1000% of its estimated value. The vertical red line indicates the replication advantage estimated during model calibration, i.e., $F_{\text{Adv}}$ = 0.32.

Our model predicts that the highest DIP amount can be produced using large quantities of both MOI and MODIP during infection (Fig 6B). However, this would require a lot of virus seed material for infection, rendering this option unattractive for large scale DIP manufacturing. Very good yields could also be achieved by lower virus input, i.e., an MOI of 0.01 and an MODIP of 0.25, which reaches over 50% of the predicted maximum DIP production using a 150 times lower seed virus concentrations for infection. In general, applying slightly higher DIP than STV concentrations during infection resulted in the best results for DIP production.

Furthermore, we investigated the effect of the intracellular parameter $F_{Adv}$, the replication advantage of DI cRNA over its FL counterpart, on infectious STV inhibition and DIP propagation. For an MODIP of $10^{-3}$, no significant impact of this parameter on infectious STV titers (Fig 6C) could be determined. When combining high STV and DIP concentrations, an increase of the parameter $F_{Adv}$ could lead to decreased infectious STV titers. A reduction of the replication advantage would in turn lead to improved STV production in this scenario. The release of DIPs could be improved when the parameter $F_{Adv}$ is increased, however, after reaching an optimal value the model predicts a decrease for higher values (Fig 6C). For equimolar virus particle concentrations, i.e., when both MOI and MODIP are either $10^{-3}$ or 30, this optimal value regarding DIP production is $F_{Adv} = 1.8$. Using MOI 30 and MODIP $10^{-3}$, $F_{Adv} = 2.5$ is predicted to be optimal for DIP production.

Taken together, the extended multiscale model of STV and DIP co-infection predicts that a MODIP-to-MOI ratio of about $10^4$:1 has to be used to restrict STV production and spreading significantly. Furthermore, an increase of the replication advantage of DI cRNAs could improve both the suppression of STV release and the production of DIPs.

## Discussion

IAV infection is an intricate biological process in which the virus and the host cell interact on multiple levels. Typically, only STV replication is considered but the influence of DIPs present at time of infection or emerging continuously adds an additional layer of complexity to the underlying system. Current experimental and computational techniques facilitate a profound investigation of such processes. In this study, we conducted a series of STV and DIP co-infection experiments and developed a mathematical multiscale model, which captures the measured intra- and extracellular dynamics closely. We used this model to predict virus propagation for a wide range of infection conditions and estimated optimal settings for STV suppression and DIP production.

Our experiments with different MOIs and MODIPs show that STV production can be reduced significantly for specific infection conditions. Additionally, the fraction of apoptotic cells for low MOI and high MODIP scenarios remained at a very low level and cells survived for over 60 hpi. Interestingly, even the generation of over $10^3$ viral mRNAs per cell and the putative translation of viral proteins in DIP-only infected cells did not lead to an augmented cell death response. Therefore, another trigger has to cause this cellular defense mechanism against infection. As discussed in [39], previous studies clearly suggested that the intracellular concentration of vRNA is a critical factor for apoptosis induction. Our experimental results support these findings and indicate that only the replication of large amounts of viral genomes in the nucleus, i.e., vRNA, will lead to virus-induced apoptosis in MDCKsus cells.

Furthermore, the wide range of applied infection conditions enables a comprehensive characterization of the effect of MOI and MODIP on virus production. In a recent study, Martin et al. [40] showed that the MOI strongly affects dynamics of STV replication in adherent MDCK cells. They observed higher virus titers and an earlier onset of virus particle release with increasing MOIs. While our experimental results of MDCKsus cell infections support a

faster release of both STVs and DIPs with higher MOIs (S3 Fig), we did not observe an overall improvement of virus titers using a larger STV input. The highest concentrations of infectious and total STV were achieved in low MOI conditions (S3A and S3B Fig). Similar results have been reported previously for adherent MDCK cells [41,42]. The maximum DIP titers in our experiments were found for an equimolar STV and DIP input (S3G and S3L Fig). This indicates that the ratio between MOI and MODIP might play a larger role than the total virus input for optimal DIP production conditions.

By measuring STV and DIP co-infection kinetics on the intracellular and cell population level for 12 different infection conditions, we obtained a multitude of experimental data, which provides a deep look into the interaction between these two viruses. However, to describe the dynamics of all observed infection conditions with a single set of parameters, we had to extend the initially developed basic multiscale model of STV and DIP co-infection by considering various steps of viral mRNA synthesis in more detail. Specifically, we introduced an internal regulation between segments, i.e., a lower transcription rate for segments encoding for RdRp-related proteins compared to structural proteins, as observed in previous replication studies [31–33]. Interestingly, for infection conditions using equal amounts of STVs and DIPs, the DI mRNA levels are exactly between FL and S5 mRNA levels (Fig 4B). This indicates that due to the replication advantage of DIPs their viral mRNA can overcome their STV counterpart, but not reach the levels of S5 and potentially other segments.

Furthermore, our experiments show that cells only infected by DIPs are able to produce high concentrations of viral mRNA for scenarios with very few co-infections, i.e., low MOI conditions. A baseline level of viral mRNA was detected, which correlates with a primary transcription mediated only by the infecting DIP. However, since segment 1 encoding for a subunit of RdRp is defective (DI244), the replication of vRNA and with that the amplification of the viral genome did not occur. A highly interesting aspect in this regard is, if cells are capable to perform vRNA replication in case that they are solely infected by DIPs with deletions in segments not encoding for RdRp. In a recent study, Phan et al. [33] investigated the levels of viral RNAs during infection for two different defective influenza viruses in A549 cells. One of them was lacking FL segment 2, which encodes for a subunit of RdRp, and showed an accumulation of viral mRNA but no replication of vRNA similar to our experimental results for condition L30. An infection with the second virus, which was lacking FL segment 4 encoding for the structural protein hemagglutinin, resulted in vRNA and viral mRNA levels similar to a wild-type infection. This indicates that the *de novo* synthesis of RdRp is critical for virus replication, while the genome-bound RdRp provided by virions entering during initial infection is sufficient for viral mRNA transcription.

The extended model was calibrated to measurements on the intracellular and cell population level simultaneously and was able to capture the observed dynamics closely (S3–S10 Figs). An important factor to capture all infection dynamics simultaneously was the introduction of an MODIP-to-MOI ratio-dependent rate of vRNA synthesis resulting in a reduction of vRNA levels in the presence of high DIP concentrations. Experimental data clearly demonstrated that higher MODIPs lead to a reduction of vRNA levels (S8 and S10 Figs). This interaction could describe a "self-interference" that has been reported previously [25,43] and was also predicted by mathematical modeling [23]. Specifically, due to high DIP levels, the viral replication is restricted and, thus, the amplification of both STVs and DIPs is affected. However, the exact mechanism of this effect and the factors involved in such a reduction cannot be clarified using a mathematical model. Our hypothesis regarding the underlying interactions is that viral replication is limited due to a strong competition for viral proteins caused by high DIP concentrations. If DI genomes occupy most RdRp, the transcription of FL mRNA could be reduced significantly and fewer functional viral proteins would be synthesized. This would ultimately

lead to a reduction of viral replication for both STVs and DIPs. To elucidate such interdependencies, further experiments focusing on the effect of high DIP levels on viral RNA replication are required.

In our model prediction, we showed that an MODIP-to-MOI ratio of at least $10^4$:1 is required to reduce STV titers by over four orders of magnitude and enable a suppression of STV infection in MDCKsus cells (Fig 6A). In line with our model prediction, recent infection experiments in mice using varying STV and DIP concentrations provided similar results [16,44]. In these studies, the complete protection induced by DIP administration was overcome when the MODIP-to-MOI ratio was reduced from $3.4 \times 10^4$:1 to $3.4 \times 10^3$:1 and from $4.4 \times 10^4$:1 to $2.2 \times 10^4$:1, respectively.

For infections in humans, about 0.6 to 3 infectious units were reported for successful airborne transmissions [44]. Extrapolating our findings from cell culture experiments, the administration of $3 \times 10^4$ DIPs, e.g. via nasal spray, could be sufficient to limit infection spread severely. However, as the preferred target tissues of IAV in humans do not correspond to a well-mixed cultivation system, the administration of higher doses is likely necessary. If we assume that the complete respiratory tract consists of about $4 \times 10^8$ cells [45] and that at least an MODIP of $10^{-1}$ should be achieved to induce strong infection suppression at such low MOIs, $4 \times 10^7$ DIPs would be required for a strong inhibiting effect. This amount of DIPs would also theoretically protect against up to 4000 infectious units, which is 1300 times the airborne infectious dose. However, if we consider an advanced infection already subject to strong virus replication, high MOI conditions could be induced. Assuming MOIs of 1 or above, at least $4 \times 10^{12}$ DIPs would be necessary to achieve a strong inhibition according to our predicted MODIP-to-MOI ratio. Most likely, the application of such high DIP doses would not be reasonable due to safety concerns. Therefore, the use of DIP preparations shows the biggest promise shortly after infection or for prophylaxis. Previous *in vivo* experiments, which administered DIPs to mice at varying times before and after infection, support this hypothesis [28].

In addition, the innate immune response induced by DIPs was shown to play a major role for their therapeutic effect [46]. This concerns, in particular, their antiviral activity against influenza B virus [47], pneumovirus [48] and SARS-CoV-2 [49]. Obviously, DIPs could also improve the defense against STVs by other means than high DI RNA replication rates and competition for intracellular resources. To evaluate such effects, a different cell line should be used. While MDCK cells show a strong interferon response following STV infection, the subsequently produced myxovirus resistance protein 1 shows a lack of activity against the human IAV due to its canine origin [50]. Moreover, it was reported that the trypsin added to the cultivation medium to facilitate the IAV entry into the cells also degrades interferon [51].

Lastly, the model predicts that for the cell culture-based production of large amounts of DIPs relatively low amounts of virus material are sufficient for infection. As long as STVs and DIPs are provided in more or less equimolar concentrations and the initial MODIP is kept above 0.1, high levels of DIPs are obtained (Fig 6B). Generally, the predicted DIP production was highest when slightly more DIPs than STVs were provided. An additional factor that should be considered for the generation of higher DIP titers in co-infections is the replication advantage of DI over FL genomes [7]. By using an optimal factor for this advantage, which is implemented as an increased synthesis rate of DI cRNA over its FL counterpart in our model, up to 3.6 times more total DIPs could be produced in simulations (Fig 6C). Furthermore, model predictions suggest a more prominent replication advantage could also improve STV titer reduction in high virus concentration scenarios. A potential strategy to obtain DIPs with higher advantages over the STV is the selection of strongly accumulating DIPs from long-term continuous bioreactor cultivations [52]. Such DIPs consistently replicated at high levels indicating an increased advantage over their competition, i.e., other emerging DIPs.

Further model extension towards description of *in vivo* infections could support the exploration of strategies to prevent virus spreading in tissues and organs. To achieve such goals, the model would require an expansion to describe virus spread in the second or third spatial dimension. Additionally, the immune system, especially the innate immune response, would need to be considered explicitly. Finally, the multiscale model developed in this study is well calibrated to optimize cell culture-based DIP production and provides a solid basis for the analysis of DIP application strategies for prophylaxis and treatment of IAV infections.

## Materials and methods

### Model of the intracellular level

The description of the intracellular dynamics of STV and DIP co-infection is based on a model developed recently in our group [23]. Briefly, this model consists of a set of ordinary differential equations that represent virus entry, viral mRNA and protein synthesis, virus genome replication, packaging of viral genomes and progeny virion release for STVs and DIPs (Eqs (S1)-(S45)). To link this model with the model established on the cell population level and to capture the infection dynamics observed in cell cultures closely, we modified various equations in the original intracellular model. For the complete set of equations see S1 Appendix.

First, we incorporated additional regulation mechanisms during viral mRNA synthesis, i.e., (I) the inhibition of mRNA transcription activity by RdRp as suggested in [53,54], and (II) a reduction of RdRp-related viral mRNA transcription as shown in [31–33]. To that end, we adjusted the viral mRNA dynamics to

$$\frac{dR_i^M}{dt} = f_M \frac{k_M^{\text{Syn}} V p_i^{\text{Nuc}}}{L_i \left(1 + \frac{P_{\text{Rdrp}}}{K_R}\right)} - k_M^{\text{Deg}} R_i^M \tag{1}$$

with

$$f_M = \begin{cases} F_M, & i \in \{1, 2, 3, 9\}, \\ 1, & i \in \{4, \dots, 8\}, \end{cases} \tag{2}$$

with $F_M$ as a reduction factor for RdRp-related viral mRNA synthesis, the concentration of unbound viral polymerase $P_{\text{RdRp}}$ and $K_R$ denoting the amount of free viral polymerase required to reduce mRNA transcription by 50%. Viral ribonucleoprotein $Vp^{\text{Nuc}}$ is the template for transcription and $L_i$ represents the length of the viral mRNA for segment i. The DI segment is referred to as segment $i = 9$. Viral mRNA synthesis and degradation rates are described by $k_M^{\text{Syn}}$ and $k_M^{\text{Deg}}$, respectively.

In our experiments, we unexpectedly observed the significant accumulation of viral mRNA in cells only infected by DIPs for low MOI, high MODIP conditions L3 and L30 (Fig 4C). To describe these dynamics, we implemented primary viral transcription events, discussed in [34–36], for DIP-only infected cells on the population level. Therefore, we assume that the viral vRNAs entering the nucleus during initial infection enable the production of large amounts of viral mRNA. However, as we used a DIP with a deletion in a genome segment related to RdRp, the replication of vRNA cannot take place (Fig 2). We implemented the primary viral mRNA transcription similar to Eq (1) as

$$\frac{dR_{i,I_{\text{DIP}}}^M}{dt} = f_M \frac{k_M^{\text{Syn}}}{L_i} \frac{D(t_I)}{C_{\text{Tot}}(t_I)} - k_M^{\text{Deg}} R_{i,I_{\text{DIP}}}^M, \quad i = 2, \dots, 9 \tag{3}$$

where $f_M$ is the reduction factor for RdRp-related viral mRNA synthesis (Eq (2)). Here, the

templates for transcription in an individual cell $\frac{D(t_I)}{C_{\text{Tot}}(t_I)}$ are the DIPs provided at the time of initial infection $t_I$. The inhibition of mRNA synthesis by RdRp is not applied in these equations, because we used a DIP containing a deletion in segment 1 encoding for PB2.

Additionally, we introduced a regulatory mechanism that affects vRNA synthesis depending on the MODIP-to-MOI ratio. Experimental results indicated a clear reduction of vRNA levels when high DIP concentrations were used for infection while no such effect could be detected when using low DIP concentrations (Fig 4E). Therefore, we modified the parameter $k_V^{\text{Syn}}$, which is used in Eq (S17) and describes the synthesis rate of vRNA, to

$$k_V^{\text{Syn}}(t_I) = \frac{K_V}{f_{\text{D,V}}(t_I)} \tag{4}$$

with a dependency on the MODIP-to-MOI ratio described by

$$f_{\text{D,V}}(t_I) = \begin{cases} \nu_1 \left( \dfrac{D(t_I)}{V(t_I)} \right)^{\nu_2}, & \dfrac{D(t_I)}{C_{\text{Tot}}(t_I)} \geq F_{\text{MODIP}}, \\[2ex] 1, & \dfrac{D(t_I)}{C_{\text{Tot}}(t_I)} < F_{\text{MODIP}}, \end{cases} , f_{\text{D,V}}(t_I) \geq 1 \tag{5}$$

where $K_V$ denotes the maximum vRNA synthesis rate and $t_I$ describes the time point at which a cell got infected. The parameters $\nu_1$ and $\nu_2$ describe the effect of the MODIP-to-MOI ratio $\frac{D(t_I)}{V(t_I)}$ on the parameter $k_V^{\text{Syn}}(t_I)$. To calculate the MODIP-to-MOI ratio, the extracellular concentrations of STV ($V(t_I)$) and DIPs ($D(t_I)$) at the time of infection are utilized. If the MODIP is above a threshold value $F_{\text{MODIP}}$ when a cell is infected, the vRNA synthesis is reduced. Based on our experiments, we determined this value to be in the range of $10^{-3}$ to 3. For model prediction in Fig 6 we assumed a value of $F_{\text{MODIP}} = 10^{-3}$.

Furthermore, the virus release kinetics were adjusted to bring them in line with the model of the population level [27] and to consider non-infectious virus particles. To enable the description of the infectious and total amount of virus particles, we introduced the FIVR $F_{\text{Par}}(\tau)$, which describes what percentage of released virions has the capacity to infect new cells. It is defined as

$$\frac{dF_{\text{Par}}}{dt} = -k_{\text{Red}}^{\text{Rel}} F_{\text{Par}} \tag{6}$$

with $k_{\text{Red}}^{\text{Rel}}$ denoting the decrease of infectious virus particle release over the life span of an infected cell. In [27] the FIVR needed to be changed depending on the infection conditions decreasing with higher MOIs. Here, we apply the same FIVR for all 12 infection conditions. Using the FIVR, we adjust the equations for infectious virions released from either STV- or co-infected cells to

$$\frac{dV_m^{\text{Rel}}}{dt} = r_{\text{STV,m}}^{\text{Rel}}(\tau) = F_{\text{Par}} k^{\text{Rel}} \frac{V_{\text{Cplx}}^{\text{Cyt}}}{V_{\text{Cplx}}^{\text{Cyt}} + D_{\text{Cplx}}^{\text{Cyt}} + K_{V^{\text{Rel}}}} \prod_j \frac{P_j}{P_j + N_{P_j} K_{V^{\text{Rel}}}} \tag{7}$$

$$\frac{dD_m^{\text{Rel}}}{dt} = r_{\text{DIP,m}}^{\text{Rel}}(\tau) = F_{\text{Par}} k^{\text{Rel}} \frac{D_{\text{Cplx}}^{\text{Cyt}}}{V_{\text{Cplx}}^{\text{Cyt}} + D_{\text{Cplx}}^{\text{Cyt}} + K_{V^{\text{Rel}}}} \prod_j \frac{P_j}{P_j + N_{P_j} K_{V^{\text{Rel}}}} \tag{8}$$

and introduce the total virus particle release

$$\frac{dV_{m,\text{Tot}}^{\text{Rel}}}{dt} = r_{\text{STV,m,Tot}}^{\text{Rel}}(\tau) = k^{\text{Rel}} \frac{V_{\text{Cplx}}^{\text{Cyt}}}{V_{\text{Cplx}}^{\text{Cyt}} + D_{\text{Cplx}}^{\text{Cyt}} + K_{V^{\text{Rel}}}} \prod_j \frac{P_j}{P_j + N_{P_j} K_{V^{\text{Rel}}}} \tag{9}$$

$$\frac{dD_{m,Tot}^{Rel}}{dt} = r_{DIP,m,Tot}^{Rel}(\tau) = k^{Rel} \frac{D_{Cplx}^{Cyt}}{V_{Cplx}^{Cyt} + D_{Cplx}^{Cyt} + K_{V^{Rel}}} \prod_j \frac{P_j}{P_j + N_{P_j} K_{V^{Rel}}} \tag{10}$$

where $j \in \{HA, NA, M1, M2\}$, $m \in \{I_{STV}, I_{CO}\}$ represents the type of cells which release virus particles, and $P_j$ denotes the available viral proteins. The amount of vRNP-complexes for either STVs ($V_{Cplx}^{Cyt}$) or DIPs ($D_{Cplx}^{Cyt}$) determines the release of progeny virions. The infection age of a cell, which represents the time that has passed since cells were infected, is described by $\tau$. The parameters $K_{V^{Rel}}$ and $N_{P_j}$ denote the amount of viral complexes necessary to achieve half the maximum virus release rate and the number of viral proteins required for the formation of virus particles, respectively. Therefore, the parameter $k^{Rel}$ acts as a maximum value for the rate of virus release. Using these equations, we can describe the dynamics of infectious virions, total STVs and total DIPs released during the infection of an animal cell culture.

## Model of the population level

The extracellular kinetics describing STV and DIP co-infection are based on a recently published multiscale model of IAV infection [27], which expanded conventional cell population dynamics by using a logistic infection age-dependent apoptosis rate. In short, a set of ordinary differential equation is coupled with integro-partial differential equations to describe infection dynamics on the cell population level. This model describes (I) growth, infection and apoptosis of uninfected cells, (II) infection-induced apoptosis of infected cells, and (III) attachment, endocytosis, production and degradation of virus particles (Eqs (S47)-(S74)). To describe the interactions of DIPs with the STV dynamics, we expanded this model by introducing DIP-related cell and virus populations. For a detailed description of the population dynamics, the reader is referred to S1 Appendix.

Based on the original model, ODEs describing the time course of uninfected target cells $T$, STV-only infected cells $I_{STV}$ and their apoptotic forms $T_A$ and $I_A$, respectively, were expanded to handle DIP-only infected $I_{DIP}$ and co-infected cells $I_{CO}$

$$\frac{dT}{dt} = \mu T - r_{STV}^{Inf} T - r_{DIP}^{Inf} T - k_T^{Apo} T \tag{11}$$

$$\frac{dI_{DIP}}{dt} = r_{DIP}^{Inf} T + \mu I_{DIP} - r_{STV}^{Inf} I_{DIP} - k_T^{Apo} I_{DIP} \tag{12}$$

$$\frac{\partial I_{STV}}{\partial t} + \frac{\partial I_{STV}}{\partial \tau} = -[r_{DIP}^{Inf} + k_I^{Apo}(\tau)]I_{STV}(t,\tau) \tag{13}$$

$$\frac{\partial I_{CO}}{\partial t} + \frac{\partial I_{CO}}{\partial \tau} = -k_I^{Apo}(\tau)I_{CO}(t,\tau) \tag{14}$$

$$\frac{dT_A}{dt} = k_T^{Apo} T - r_{STV}^{Inf} T_A - r_{DIP}^{Inf} T_A - k^{Lys} T_A \tag{15}$$

$$\frac{dI_A}{dt} = \int_0^\infty k_I^{Apo}(\tau)[I_{STV}(t,\tau) + I_{CO}(t,\tau)]d\tau + k_T^{Apo} I_{DIP} + r_{STV}^{Inf} T_A + r_{DIP}^{Inf} T_A - k^{Lys} I_A \tag{16}$$

$$C_{Tot}(t) = T(t) + T_A(t) + I_{DIP}(t) + \int_0^\infty I_{STV}(t,\tau)d\tau + \int_0^\infty I_{CO}(t,\tau)d\tau + I_A(t) \tag{17}$$

with

$$\mu(t) = \left(f_\mu \frac{\mu_{\text{Max}}}{T_{\text{Max}}}[T_{\text{Max}} - C_{\text{Tot}}(t)]\right)_+ \tag{18}$$

and

$$f_\mu = \begin{cases} F_\mu, & [V(0) + D(0)]C_{\text{Tot}}(0)^{-1} \geq 6, \\ 1, & [V(0) + D(0)]C_{\text{Tot}}(0)^{-1} < 6, \end{cases} \tag{19}$$

Uninfected and DIP-only infected cells get apoptotic with the same rate $k_T^{\text{Apo}}$ and grow with the specific rate $\mu$, with a maximum value $\mu_{\text{Max}}$. This specific rate is affected by very high virus concentrations during infection via the factor $F_\mu$. While suspension cell growth is generally not restricted severely by the available space in a vessel, we utilized the maximum cell concentration $T_{\text{Max}} = 10^7$ cells/mL measured in our experiments as an upper limit. The infection of cells by STVs and DIPs is described by the rates $r_{\text{STV}}^{\text{Inf}}$ and $r_{\text{DIP}}^{\text{Inf}}$, respectively. Target cells and their apoptotic counterpart can get infected by either STVs or DIPs, however, re-infection of STV- and DIP-only infected cells is only possible by the opposing virus particle. Additionally, STV-only infected cells are protected from re-infection after reaching an infection age of 3 h to consider superinfection exclusion, which is mediated by neuraminidase [55,56]. Following the implementations in [29] and [27], the infection age $\tau$ of STV- and co-infected cells is considered and both populations undergo apoptosis with an infection age-dependent apoptosis rate $k_I^{\text{Apo}}(\tau)$. Cell lysis of apoptotic target and apoptotic infected cells is described by the rate $k^{\text{Lys}}$.

Furthermore, we added DIPs on the population level following the description of STVs and defined them as

$$\frac{dD}{dt} = \int_0^\infty [r_{\text{DIP,I}_{\text{STV}}}^{\text{Rel}}(\tau)I_{\text{STV}}(t,\tau) + r_{\text{DIP,I}_{\text{CO}}}^{\text{Rel}}(\tau)I_{\text{CO}}(t,\tau)]d\tau - k_D^{\text{Deg}}D + \sum_n(k_n^{\text{Dis}}D_n^{\text{Att}} - k_{c,n}^{\text{Att}}B_n^D D) \tag{20}$$

with $r_{\text{DIP,m}}^{\text{Rel}}(\tau)$ as the age-dependent DIP release rate of $m \in \{I_{\text{STV}}, I_{\text{CO}}\}$ cells. The age-segregated cell populations $I_{\text{STV}}(t,\tau)$ and $I_{\text{CO}}(t,\tau)$ can both produce DIPs and degradation occurs with a rate of $k_D^{\text{Deg}}$. The dissociation and association of DIPs from cells is described by $k_n^{\text{Dis}}$ and $k_n^{\text{Att}}$ with $n \in \{\text{hi, lo}\}$, respectively. $B_n^D$ refers to the amount of virus binding sites on the cell surface to which DIPs can attach. DIPs attached to cells ($D_n^{\text{Att}}$) and inside cellular endosomes ($D^{\text{En}}$) were implemented analogous to the corresponding STV versions (Eqs (S66)-(S69)).

## Simulation approach and parameter estimation

Generally, model simulation was performed based on previously published multiscale models [27,29]. However, the intracellular and population model are not decoupled anymore, because we assume that the extracellular level has an impact on intracellular events. As before, the intracellular and population models are linked by the virus release rates, i.e., $r_{\text{STV}}^{\text{Rel}}$ and $r_{\text{DIP}}^{\text{Rel}}$. These rates are calculated on the intracellular level depending on the infection age $\tau$ and determine virus release on the population level (Eqs (S42)-(S45)). In addition, we assume that the current number of STVs ($V(t)$, $V_n^{\text{Att}}(t)$, $V^{\text{En}}(t)$) and DIPs ($D(t)$, $D_n^{\text{Att}}(t)$, $D^{\text{En}}(t)$) on the extracellular level dictates the initial conditions for cells infected at this specific time $t$. In contrast to the previous approaches, we did not utilize a reduced intracellular model and simulated the intracellular model directly based on the state of the population level. This change increased computational burden considerably, however, it also enables the representation of infections with highly dynamic virus concentrations.

The intracellular model (Eqs (S1)-(S45)) was solved numerically using the CVODE routine from SUNDIALS [57] on a Linux-based system. The Systems Biology Toolbox 2 [58] was employed in MATLAB (version 9.2.0.556344, R2017a) to process model files and experimental data. The population model (Eqs (S47)-(S74)) and Eq (S46) were calculated using Euler's method with a step size $dt$ = 0.1 h. In case this step size lead to rapidly oscillating behavior, e.g. when the concentration of uninfected target cells reached values close to zero, it was reduced to $dt$ = 0.02 h. The integrals in Eqs (S55)-(S56), (S59)-(S60), (S62), (S64)-(S65), (S71) and (S73) were calculated by substituting Eq (S51) for $I_{STV}(t, \tau)$, Eq (S52) for $I_{CO}(t, \tau)$ and applying the rectangle rule to approximate results.

As in [27], we assume that cells are infected in the moment a virus genome enters their cytoplasm and that at least one complete STV or DIP is required for infection. Therefore, the initial values for $V^{Cyt}(0)$ and $D^{Cyt}(0)$, which describe the amount of STVs and DIPs in the cytoplasm before nuclear import, are set to 1 for simulation of the intracellular model. The other initial conditions, i.e., viral species $V^{Ex}(0)$, $V_n^{Att}(0)$, $V^{En}(0)$, $D^{Ex}(0)$, $D_n^{Att}(0)$ and $D^{En}(0)$, are taken from the current state of the population model. For simulation of cells only infected by STVs, all DIP-related initial values are set to 0.

Furthermore, we assume that the minimum release from an infected cell is one complete virus particle. To that end, all values in the infection age-dependent release rates $r_{STV}^{Rel}(\tau)$, $r_{STV,Tot}^{Rel}(\tau)$, $r_{DIP}^{Rel}(\tau)$, and $r_{DIP,Tot}^{Rel}(\tau)$ that are below 1 are set to 0. This introduces a certain delay between the infection of cells and the subsequent release of virions, which prevents an unreasonably rapid virus spread in short time intervals, especially for low MOI conditions.

For parameter calibration, the intracellular and the population model were fitted simultaneously to experimental data from all 12 infection conditions (S1 and S3–S10 Figs). On the intracellular level, we obtained vRNA and viral mRNA measurements. Cell population dynamics and viral titers for infectious STVs, total STVs and total DIPs were determined on the extracellular level. To estimate a single set of parameters, which enables the description of all infection conditions at the same time, we used the evolutionary optimization algorithm CMA-ES [59]. During model calibration, intermediate estimation results were assessed by normalizing errors to their respective maximum measurement value. Then, the SSRs determined on the intracellular and population level were divided by the corresponding number of data points and added to evaluate the quality of fits. For simulated values of vRNAs and viral mRNAs, the first measured data point was added as an offset to accommodate for a background signal in the real-time RT-qPCR analysis.

The final parameter values are presented in S2 Table and S3 Table. An overview of the local sensitivity for all model parameters, which was calculated based on Heldt et al. [60], is provided in S4 Table. The confidence intervals shown in Table 1 were calculated using a bootstrapping method [38] considering a standard deviation of 55% for qPCR measurements and a standard deviation of 40% for cell population data based on experiments from [61]. Additionally, an error of 40% for the infectious STV titer was applied based on test runs using the PFU assay.

## Model prediction

For the prediction of infectious STV and total DIP release for various infection conditions (Fig 6), we simulated the extended multiscale model using the parameters calibrated to our experimental data. The amount of initially available STVs and DIPs on the population level was adjusted to the intended values by multiplying the corresponding MOI and MODIP with a viable cell concentration of $T(0) = 2.2 \times 10^6$ cells/mL. The values in Fig 6 show the maximum concentration of progeny STVs and DIPs on the population level until 48 hpi.

To evaluate the impact of the replication advantage of DI cRNA on STV and DIP release, we simulated the multiscale model using adjusted parameter values (Fig 6C). Therefore, we varied the parameter $F_{Adv}$ between 0 and 1000% of its estimated value and performed model

simulations for different MOI and MODIP conditions. The fold-change shown in Fig 6C was calculated by comparing the resulting virus titers at 24 hpi with simulation outcomes obtained by using the unmodified parameter $F_{Adv} = 0.32$.

## Cells and viruses

An adherent Madin-Darby Canine Kidney (MDCK) cell line (ECACC, No. 84121903), adapted first to growth in suspension [62] and subsequently to growth in the chemically defined medium Xeno [63], in the following referred to as MDCKsus cells, was used. The medium was supplemented with 8 mM glutamine. Cells were cultivated in shake flasks (125 mL baffled Erlenmeyer Flask, Thermo Fisher Scientific, 4116–0125) at a working volume of 50 mL in an orbital shaker (Multitron Pro, Infors HT; 50 mm shaking orbit) at 185 rpm, 37°C, 5% $CO_2$. The parental adherent MDCK cells ("MDCKadh", ECACC, No. 84121903) used for determination of infectious STV titers (PFU/mL, see below) were cultured in Glasgow minimum essential medium (GMEM, Thermo Fisher Scientific, #221000093) containing 1% peptone and 10% fetal bovine serum at 37°C and 5% $CO_2$.

For STV infection, an influenza A virus strain A/PR/8/34 of subtype H1N1 (PR8) (provided by Robert Koch Institute, Berlin, Germany), adapted to MDCKsus cells and Xeno medium [63] was used. Infectious virus titer ($0.8 \times 10^9$ $TCID_{50}$/mL) of the seed virus was quantified via $TCID_{50}$ assay [30]. Generation of purely clonal DI244 was conducted according to [64] and production as specified in [15]. "Active DIP titer" of the seed virus ($1.5 \times 10^8$ PFU/mL) was determined as described in [15]. Depletion of DIPs in the seed virus was controlled by segment-specific PCR according to [20,65].

## Infection

MDCKsus cells were infected with different STV doses (corresponding to MOIs of $10^{-3}$, 3, 30) and DI244 doses (MODIPs of 0, $10^{-3}$, 3, 30), as shown in Fig 1. We calculated the MOIs based on the $TCID_{50}$ titer, while MODIPs were calculated based on the "active DIP titer" as described in [15]. For infection, we added trypsin at a final activity of 20 U/mL. After inoculation with virus, cells were washed at 0.75 hpi with pre-warmed phosphate-buffered saline (PBS) ($300 \times g$, 5 min, room temperature), and cells were provided with fresh infection medium containing trypsin for subsequent cultivation.

## Sampling for analytics

For sampling at indicated time points post infection, viable cell concentration was measured via a cell counter (Vi-Cell XR, Beckman coulter, #731050). Next, $1 \times 10^6$ cells were centrifuged ($300 \times g$, 5 min, 4°C), and supernatant was discarded. Cell pellets were lysed with 350 μL lysis buffer "RA1" (from "NucleoSpin RNA" kit, Macherey-Nagel, 740955) supplemented with 1% (v/v) β-mercaptoethanol and stored at -80°C until real-time RT-qPCR analytics. In addition, aliquots of cell suspensions were centrifuged ($300 \times g$, 5 min, 4°C) and supernatants were stored at -80°C until virus titration or real-time RT qPCR analytics. The remaining cell pellet was fixed with paraformaldehyde (1% (w/v)), and processed according to a previously published protocol for cell sampling required for imaging flow cytometry analysis [61,65].

## Plaque assay

Quantification of the "active DIP titer" of the DI244 seed virus by plaque assay followed an established protocol [15]. Furthermore, the plaque assay using MDCKadh cells was applied to determine the course of STV titers.

### Real-time RT-qPCR

vRNAs in supernatants were isolated using the "NucleoSpin RNA virus" kit (Macherey-Nagel, 740956), and vRNAs in cell pellets utilizing the "NucleoSpin RNA" kit (Macherey-Nagel, 740955) according to the manufacturers' instructions. A previously published method [61,65,66] was used for the absolute quantification of vRNAs and mRNAs using real-time reverse transcription qPCR (real-time RT-qPCR). Primers for reference standard generation and specific detection of FL segment 1 and DI244 vRNA [67], and S5 vRNA and viral mRNA [61,65] were used. Primers of FL segment 1 and DI244 vRNA and viral mRNA are listed in S5 Table and S6 Table.

### Imaging flow cytometry

An established protocol for imaging flow cytometric analysis of cells was utilized [61,65]. In brief, cells were stained for NP using a monoclonal mouse anti-NP mAb61A5 (provided by Fumitaka Momose) at a dilution of 1:100 and a secondary antibody Alexa Fluor 647-conjugated polyclonal goat anti-mouse (Thermo Fisher, #A21235) at a dilution of 1:500. DAPI was added for nuclear staining. Acquisition of 10,000 single cells for each sample was performed using the ImageStream X Mark II (Luminex). For data analysis, IDEAS software was utilized. vRNP positive cells (infected cells) were determined based on a gate set on mock infected cells (1% threshold). Apoptotic cells were detected based on image analysis, evaluating chromatin condensation, nuclear fragmentation and cell shrinkage [61,65,68]. Furthermore, we determined fractions of the whole cell population that were (I) infected and apoptotic, (II) infected and non-apoptotic, (II) non-infected and apoptotic, and (IV) non-infected and non-apoptotic [27,29].

### Supporting information

**S1 Appendix. Full list of equations for the multiscale model.**
(DOCX)

**S1 Data. Experimental data used for model calibration.**
(XLSX)

**S1 Fig. Cell concentration and fraction of apoptotic cells for all MOI and MODIP conditions.** Measurements of (A-C) viable cell concentration and (D-F) the fraction of apoptotic cells for infections with MOI $10^{-3}$, 3 and 30 using different MODIPs.
(TIF)

**S2 Fig. Model extension significantly improves description of experimental measurements.** The sum of squared residuals for each individual measured property is depicted. Logarithmic errors of each variable were normalized to the respective maximum measurement value. The (A) basic model and the (B) extended model were calibrated to a wide range of experimental data. Measured properties include vRNA and mRNA of full-length (FL) segment 1, defective-interfering (DI) segment 1 and segment 5 (S5), the concentration of uninfected, infected and apoptotic cells, total and standard virus (STV) titers as well as DIP titers.
(TIF)

**S3 Fig. Experimental data and model simulations for virus titers.** Model fits to measurements of the infectious STV titer, the total amount of STVs and the total amount of DIPs for MDCKsus infections with MOI $10^{-3}$, 3 and 30 using different MODIPs.
(TIF)

**S4 Fig. Experimental data and model simulations for cell populations.** Model fits to measurements of the fraction of uninfected, uninfected and apoptotic, infected, infected and apoptotic cells for MDCKsus infections with MOI $10^{-3}$, 3 and 30 using different MODIPs.
(TIF)

**S5 Fig. Experimental data and model simulations for FL mRNA dynamics.** Model fits to measurements of the intracellular levels of FL mRNA for MDCKsus infections with MOI $10^{-3}$, 3 and 30 using different MODIPs.
(TIF)

**S6 Fig. Experimental data and model simulations for DI mRNA dynamics.** Model fits to measurements of the intracellular levels of DI mRNA for MDCKsus infections with MOI $10^{-3}$, 3 and 30 using different MODIPs.
(TIF)

**S7 Fig. Experimental data and model simulations for segment 5 mRNA dynamics.** Model fits to measurements of the intracellular levels of segment 5 mRNA for MDCKsus infections with MOI $10^{-3}$, 3 and 30 using different MODIPs.
(TIF)

**S8 Fig. Experimental data and model simulations for FL vRNA dynamics.** Model fits to measurements of the intracellular levels of FL vRNA for MDCKsus infections with MOI $10^{-3}$, 3 and 30 using different MODIPs.
(TIF)

**S9 Fig. Experimental data and model simulations for DI vRNA dynamics.** Model fits to measurements of the intracellular levels of DI vRNA for MDCKsus infections with MOI $10^{-3}$, 3 and 30 using different MODIPs.
(TIF)

**S10 Fig. Experimental data and model simulations for segment 5 vRNA dynamics.** Model fits to measurements of the intracellular levels of segment 5 vRNA for MDCKsus infections with MOI $10^{-3}$, 3 and 30 using different MODIPs.
(TIF)

**S11 Fig. The basic model fails to describe virus replication and propagation dynamics for all infection conditions.** Curves represent model simulations of the basic model calibrated to (A-C) cell-specific vRNA, (D-F) cell-specific viral mRNA, (G-I) cell population and (J-L) virus titer data measured in MDCK suspension cell cultures infected with different amounts of influenza A/PR/8/34 (H1N1) and defective interfering particles (DI244). Results from MOIs and MODIPs of $10^{-3}$ and 0 (first column), 30 and 3 (second column), $10^{-3}$ and 30 (third column) are shown. The basic model describes IAV and DIP replication and propagation based on Rüdiger et al. [5] and Laske et al. [1] without considering additional model adaptations.
(TIF)

**S1 Table. Evaluation of the model fits performed for the basic and the extended model for individual infection conditions.**
(DOCX)

**S2 Table. Parameters of the intracellular model.**
(DOCX)

**S3 Table. Parameters of the cell population model.**
(DOCX)

**S4 Table. Sensitivity of intracellular and cell population model parameters.**
(DOCX)

**S5 Table. Primers used for real-time RT qPCR of mRNA.**
(DOCX)

**S6 Table. Primers used for reference standard generation of mRNA.**
(DOCX)

## Acknowledgments

We thank Nancy Wynserski and Claudia Best for excellent technical assistance and Tanja Laske for critical comments on the manuscript.

## Author Contributions

**Conceptualization:** Daniel Rüdiger, Lars Pelz, Sascha Y. Kupke, Udo Reichl.

**Data curation:** Lars Pelz, Marc D. Hein.

**Formal analysis:** Daniel Rüdiger, Lars Pelz, Marc D. Hein.

**Funding acquisition:** Udo Reichl.

**Investigation:** Daniel Rüdiger, Lars Pelz, Marc D. Hein.

**Methodology:** Daniel Rüdiger, Lars Pelz, Marc D. Hein, Sascha Y. Kupke.

**Project administration:** Sascha Y. Kupke, Udo Reichl.

**Software:** Daniel Rüdiger.

**Supervision:** Sascha Y. Kupke, Udo Reichl.

**Validation:** Daniel Rüdiger, Lars Pelz, Sascha Y. Kupke.

**Visualization:** Daniel Rüdiger, Lars Pelz.

**Writing – original draft:** Daniel Rüdiger, Lars Pelz, Sascha Y. Kupke.

**Writing – review & editing:** Daniel Rüdiger, Lars Pelz, Marc D. Hein, Sascha Y. Kupke, Udo Reichl.

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
