## [Decision Letter · Decision Letter 0]

2 Jun 2021

Dear Mr. Rüdiger,

Thank you very much for submitting your manuscript "Multiscale model of defective interfering particle replication for influenza A virus infection in animal cell culture" for consideration at PLOS Computational Biology. As with all papers reviewed by the journal, your manuscript was reviewed by members of the editorial board and by several independent reviewers. The reviewers appreciated the attention to an important topic. Based on the reviews, we are likely to accept this manuscript for publication, providing that you modify the manuscript according to the review recommendations.

Sincerely,

Jason M. Haugh

Deputy Editor

PLOS Computational Biology

Jason Haugh

Deputy Editor

PLOS Computational Biology

[LINK]

Reviewer's Responses to Questions

**Comments to the Authors:**

Reviewer #1: Review uploaded as attachment.

Reviewer #2: The work by Rudiger et al. investigates the dynamic impacts of a defective interfering particle (DIP) of influenza A virus (IAV) on the standard IAV virus (STV) using a combination of experimental and mathematical modeling approach. The authors infected MDCK cells suspension cells using 12 different combinations of MOIs for STV and DIPs (MODIP is used to denote MOIs for DIPs), and measured a variety of quantities of interests including levels of cell deaths, vRNA, viral mRNA, total virus yield and infectious virus yield. This data set suggests that DIP can effectively inhibit STV replication and prevent cell death when the ratio of MODIP over MOI is very high (e.g. >1000). The authors further developed a multiscale mathematical model considering viral replication in a cell, and viral spread between cells. This model is an extension of models developed previously by the authors. Fitting the model to the experimental data, the authors identified key intracellular mechanisms that regulate transcription of different gene segments of IAV and the DIP. Finally, model simulations revealed conditions under which DIP would be able to completely inhibit STV growth in cell culture and conditions allowing for maximum DIP production.

Overall, I found this work very interesting, well designed and executed. First of all, DIP has been proposed to be a promising therapeutic approach against novel viral outbreaks. Understanding the dynamic impacts of DIPs on STV is key to the development of this approach into clinical use. Therefore, this work addresses an important question. Second, the data presented here are novel in that it explores how DIP impacts on STV under different MOIs. A high MOI is required for the survival of DIP and thus the ability of DIP to inhibit STV. The works provides a perfect dataset to quantify how effectiveness of DIP is quantitatively related to MOI. Third, the application of the multiscale model framework to understand the datasets is well designed and executed. Although the model is very complicated and it is not clear to me whether such a complicated model is absolutely necessary, using such a model is extremely useful given the highly nonlinear interactions between DIP and STV and the highly complex patterns in the dataset. Lastly, the work is well written. Therefore, I applaud the authors effort to perform the interesting experiments and using the multiscale framework to interpret the patterns of the datasets.

I do not have any major concerns; here are some minor concerns that need to be addressed. First, the conclusions from the model development and model fitting section seem to be made based on visual inspections, rather than rigorous statistical tests (at least from what is written). I think it will be important to do or present quantitative analyses of the model (especially for PLOS CB). For example, it would be useful to know how many ODEs and parameter values there are in the multiscale model. The goodness of fits or AIC values of the basic model and extended model.

Second, I think it would be interesting and important to test how sensitive the fit of the model to data with respect to changes in the fixed parameter values in the model. This will allow one to see which part of the (complicated) model is important in explaining the data.

Third, it would be interesting to put the results of the works, e.g. the role of MOI on effectiveness of DIP, into the context of existing literature (in the Discussion perhaps). For example, the recent work by Martin et al. (https://doi.org/10.1371/journal.ppat.1008974) showed MOI of the STV has large impacts on the dynamics of STV replication and the interferon responses. It would be interesting to know how the data/results of the work here is related to Martin et al.

**Have the authors made all data and (if applicable) computational code underlying the findings in their manuscript fully available?**

Reviewer #1: Yes

Reviewer #2: None

PLOS authors have the option to publish the peer review history of their article (what does this mean?). If published, this will include your full peer review and any attached files.

Reviewer #1: No

Reviewer #2: **Yes: **Ruian Ke

Figure Files:

Data Requirements:

Reproducibility:

References:

---

## [Decision Letter · Decision Letter 1]

18 Aug 2021

Dear Mr. Rüdiger,

We are pleased to inform you that your manuscript 'Multiscale model of defective interfering particle replication for influenza A virus infection in animal cell culture' has been provisionally accepted for publication in PLOS Computational Biology.

Best regards,

Rustom Antia

Associate Editor

PLOS Computational Biology

Jason Haugh

Deputy Editor

PLOS Computational Biology

Reviewer's Responses to Questions

**Comments to the Authors:**

Reviewer #1: This article is investigates the relationship and dynamics of the standard influenza A virus (STV) with the defective interfering particle (DIP). MDCKsus cell lines were infected with different combinations of doses of STV and DIP and total cell death, total virus levels, total infectious virus levels, viral mRNA and other quantities of interest were measured. Experimental methods and results were thoroughly explained and included the caveat that other cell lines may have slightly different results. Findings from these experiments suggest DIPs can interfere with STV replication in accordance with other studies. More specifically, these experimental findings suggest that relatively high ratios of MODIPs (103 or 1030) to MOI (10-3) led to under 20% apoptosis despite the high DIP infection levels and also reduced total viral titers.

This experimental data was then utilized in their mathematical model, based off of the authors’ previously published work, and a model extension meant to be usable in many different dose cases. This multi scale model considers both the intracellular and cell population effects of DIP infection, STV infection, or STV and DIP co-infection. The extended version of the model included additional modifications to viral mRNA kinetics for DIP-only infected cells, a parameter for mRNA transcription that included DI segments, a vRNA synthesis parameter dependent on the MODIP to MOI ratio during infection, and a factor for cell growth rate dependent on the initial DIP concentration. This extended model is shown to fit more of the dose combination scenarios both qualitatively and, with the AIC, quantitatively in a well explained and thoughtful manner. With the solid fits and parameter estimates, the extended model was then used to predict optimal infection conditions to use DIPs to suppress STV infection.

This paper not only introduces new data on the dynamics and interaction of DIPs with STV infections but also suggests a useful mathematical model to further explore this developing therapeutic area. The extended model may have added complexity but as shown fits a wider range of MOI and MODIP infection doses with better AIC scores and the reasoning for the extensions were all well supported. All of my minor concerns were addressed and I found that any further changes made by the authors only aided in clarity of the conclusions and provided even more relevant background/publications. This is interesting and engaging look at DIPs both through experimentation and multiscale modeling.

Reviewer #2: All my concerns are address. This is a nice piece of work!

**Have the authors made all data and (if applicable) computational code underlying the findings in their manuscript fully available?**

Reviewer #1: Yes

Reviewer #2: None

PLOS authors have the option to publish the peer review history of their article (what does this mean?). If published, this will include your full peer review and any attached files.

Reviewer #1: **Yes: **Ariel Nikas

Reviewer #2: **Yes: **Ruian Ke

---

## [Editor Report · Acceptance letter]

30 Aug 2021

PCOMPBIOL-D-21-00614R1 

Multiscale model of defective interfering particle replication for influenza A virus infection in animal cell culture

Dear Dr Rüdiger,

I am pleased to inform you that your manuscript has been formally accepted for publication in PLOS Computational Biology. Your manuscript is now with our production department and you will be notified of the publication date in due course.

With kind regards,

Zsofi Zombor
